# The Use of the Shikani Video-Assisted Intubating Stylet Technique in Patients with Restricted Neck Mobility

**DOI:** 10.3390/healthcare10091688

**Published:** 2022-09-04

**Authors:** Tung-Lin Shih, Ker-Ping Koay, Ching-Yuan Hu, Hsiang-Ning Luk, Jason Zhensheng Qu, Alan Shikani

**Affiliations:** 1Department of Anesthesia, Hualien Tzuchi Hospital, Hualien 97002, Taiwan; 2Bio-Math Laboratory, Department of Financial Engineering, Providence University, Taichung 43301, Taiwan; 3Department of Anesthesia, Critical Care and Pain Medicine, Massachusetts General Hospital, Harvard Medical School, Boston, MA 02114, USA; 4Division of Otolaryngology-Head and Neck Surgery, LifeBridge Sinai Hospital, Baltimore, MD 21218, USA; 5Division of Otolaryngology-Head and Neck Surgery, MedStar Union Memorial Hospital, Baltimore, MD 21218, USA

**Keywords:** airway management, difficult airway, endotracheal intubation, Shikani technique, video-assisted intubating stylet, laryngoscopy, videolaryngoscopy, cervical spine, stereotactic, neurosurgery

## Abstract

Among all the proposed predictors of difficult intubation defined by the intubation difficulty scale, head and neck movement (motility) stands out and plays as a crucial factor in determining the success rate and the degree of ease on endotracheal intubation. Aside from other airway tools (e.g., supraglottic airway devices), optical devices have been developed and applied for more than two decades and have shown their superiority to conventional direct laryngoscopes in many clinical scenarios and settings. Although awake/asleep flexible fiberoptic bronchoscopy is still the gold standard in patients with unstable cervical spines immobilized with a rigid cervical collar or a halo neck brace, videolaryngoscopy has been repeatedly demonstrated to be advantageous. In this brief report, for the first time, we present our clinical experience on the routine use of the Shikani video-assisted intubating stylet technique in patients with traumatic cervical spine injuries immobilized with a cervical stabilizer and in a patient with a stereotactic headframe for neurosurgery. Some trouble-shooting strategies for this technique are discussed. This paper demonstrates that the video-assisted intubating stylet technique is an acceptable alternative airway management method in patients with restricted or confined neck motility.

## 1. Introduction

Direct laryngoscopy, the most commonly employed method of tracheal intubation, creates some degree of cervical spine extension while aligning the oropharyngeal and laryngeal axes. The sniffing position, traditionally used during tracheal intubation, involves the near-full extension of the atlanto-occipital and atlanto-axial joints and the flexion of the lower cervical spine [1]. This may present an issue during the endotracheal intubation of trauma patients with cervical spine (C-spine) injuries and an unstable cervical spine, even more so if the situation is also complicated by a difficult airway. A variety of intubation devices have been used to assist in the direct endotracheal tube placement with minimal manipulation of the cervical spine. Videolaryngoscopy (VL) using a GlideScope has been proposed to be safer and more effective than direct laryngoscopy (DL) using a Macintosh blade in patients with limited C-spine mobility [2,3,4,5,6,7,8,9,10,11,12]. The GlideScope has some innate shortcomings, however; it still often involves some degree of C-spine movement, and the intubation time when using the GlideScope is longer than that when using DL, i.e., 53 s vs. 24 s [13]. Other airway devices have also been successfully used in patients with restricted or confined C-spine injuries, including supraglottic airway devices, gum elastic bougies, and rigid endoscopes [14,15,16]. Flexible fiberoptic laryngoscopes and/or bronchoscopes remained as the gold standard for suspected or confirmed C-spine injuries for many years [17,18,19,20,21]. The pros and cons of the available airway management tools for patients with a potential cervical spine injury have been reviewed [1]. It was not until the introduction of optical stylets in the 1980s and 1990s that newer and safer methods of intubation were introduced, including the Shikani optical stylet and the Bonfils stylet, which are associated with a significantly lesser extension at C1–C2 and C2–C3, as compared to intubation by direct laryngoscopy [16,22,23,24,25,26].

Our institutions have extensive experience with the Shikani optical stylet, which has been widely studied in the management of a variety of difficult airways [27,28,29,30] and which has also proved effective in clinical conditions related to C-spine motility [31,32,33,34,35,36,37,38,39,40,41]. In this report, we present our experience with the Shikani video-assisted intubating stylet technique (Figure 1) as the routine airway management modality for patients with restricted head/neck mobility.

## 2. Case Presentations

**Case 1** (multiple-trauma case with rigid cervical collar immobilization): A 30-year-old female (height 170 cm, weight 68 kg, BMI 23.5 kg/m^2^) with no underlying disease was admitted to the emergency department due to multiple trauma. She fell from a 70 m-high cliff while climbing a mountain two days before. She sustained a right ankle open fracture with dislocation, a 5th cervical spine lamina fracture, a 3rd lumbar spine fracture, a left distal radius fracture, and a right ulnar shaft fracture. She was taken to the operating room for emergency external skeletal fixation (ESF) with debridement of the right ankle open fracture. The patient had a cervical collar that limited the neck mobility, and mouth opening was thee fingers wide (Figure 2). Minimal movement of the cervical spine during airway intervention and positioning of the patient to avoid further spine injury were made with caution. Standard vital-sign monitoring with electrocardiography (ECG), arterial blood pressure, pulse oximeter (SpO_2_), peripheral nerve stimulator (ToF), and end-tidal CO_2_ (EtCO_2_) were applied. The modified rapid sequence induction technique was adopted with adequate pre-oxygenation, glycopyrrolate (0.2 mg), lidocaine (20 mg), propofol (2.5 mg/kg), and high-dose rocuronium (1.0 mg/kg). Cricoid pressure was not applied during apneic face mask oxygenation. Endotracheal intubation was performed using the video-assisted intubating stylet technique. The airway was then secured swiftly and accurately (from the lip to the trachea, 7 s, Figure 3). Anesthesia was maintained with inhalational anesthetic and rocuronium. The whole anesthesia was uneventful, and the patient was extubated post-operatively.

**Case 2** (multiple trauma with rigid cervical collar immobilization): A 71-year-old female patient (height 160 cm, weight 70 kg, BMI 27.3 kg/m^2^, ASA class IIE) presented to the emergency department due to a traffic accident. Her past medical history included severe osteoporosis, neck pain, and left arm paresthesia. A radiographic work-up revealed fractures of the left distal tibial and fibular shafts. No apparent fracture of the cervical spine was noted, but severe degenerative joint disease with neuroforaminal stenosis and spinal nerve root impingement at the 5th/6th cervical vertebrae were present. The patient was scheduled for left ankle open reduction and internal fixation. She presented to the operating room with a cervical collar, and the range of motion of the neck was limited (Figure 4). The patient opened her mouth with limited vertical opening (two fingers wide). Minimal movement of the cervical spine during airway intervention and positioning of the patient was made with caution to avoid further spine injury. Standard vital-sign monitoring with electrocardiography (ECG), arterial blood pressure, pulse oximeter (SpO_2_), peripheral nerve stimulator (ToF), and end-tidal CO2 (EtCO_2_) were applied. General anesthesia was induced with glycopyrrolate, lidocaine, fentanyl, propofol, and rocuronium. She was intubated using a video-assisted intubating stylet (Figure 5). It took 40 s (from the lip to the trachea) to complete the intubating process without the jaw-thrust maneuver.

**Case 3** (stereotactic neurosurgery with deep-brain stimulation): A 57-year-old male (height 169.9 cm, weight 92.7 kg, BMI 32.0 kg/m^2^) with a history of hypertension, type 2 diabetes mellitus, hyperlipidemia, gout, and Parkinson’s disease was scheduled to undergo a stereotactic implantation of deep-brain stimulation electrodes under general anesthesia. A Leksell G-frame (Elekta Instrument Inc., Norcross, GA, USA) was applied for the stereotactic procedure. The head frame was secured on a Mayfield adaptor, and the target coordinates were applied to the stereotactic frame and the working stage. The head frame for stereotactic neurosurgery allows the precise localization of the desired target in the central nervous system. Such frame mounting is performed under local anesthesia. After mounting the frame in several potential clinical scenarios, magnetic resonance imaging (MRI) and stereotactical computed tomography (CT) were carried out. The patient was then sent to the operating room with a stereotactic frame on the head (Figure 6). Standard monitoring with electrocardiography (ECG), arterial blood pressure (NIBP), pulse oximeter (SpO_2_), peripheral nerve stimulator (ToF), and end-tidal CO_2_ (ETCO_2_) were applied. General anesthesia was induced with fentanyl, propofol, and rocuronium. Endotracheal intubation using DL would not have been possible due to the head frame. Intubation was hence performed using a video-assisted intubating stylet (Figure 6 and Figure 7). An assistant facilitated the intubating procedure with the jaw thrust maneuver. It took 20 s (from the lip to the trachea) to complete the intubating process. The anesthesia was maintained with remifentanil or alfentanil and sevoflurane. The entire deep-brain stimulation (DBS) procedure lasted for 5 h. The patient was extubated at the end of the procedure. The details of the endotracheal intubation in the presence of a stereotactic headframe are shown in Figure 6 and Figure 7 and in the Appendix A.

## 3. Discussion

In this case series, we demonstrated the use of the Shikani video-assisted intubating stylet technique in patients with restricted neck extension. The Shikani Optical Stylet (SOS; Clarus Medical, Minneapolis, MN, USA) was firstly evaluated in 1999 in a cohort of 120 adults and children scheduled for ENT procedures [24]. It has been extensively used since and proved to be invaluable in the management of a variety of difficult airways. It has also been reported that it is associated with less cervical spine motion than direct laryngoscope (DL) using a Macintosh blade [26]. The Shikani stylet was found to be safe and effective in patients undergoing surgery for cervical spondylosis, when compared with Macintosh laryngoscopy, especially beneficial in the setting of a difficult airway [36]. Awake tracheal intubation with the Shikani stylet was found as equally successful and faster as awake intubation with a flexible fiberscope in patients with unstable cervical spine [36]. To our knowledge, this is the first case report in the literature describing the use of optical stylets during stereotactic brain surgery (Figure 6 and Figure 7). The stereotactic headframe used to stabilize the head and neck during stereotactic brain surgery makes accessing the airway a challenge, and conventional DL is of limited use in such scenarios [42,43]. Few studies proposed alternative airway management modalities, including using a flexible FOB, an intubating laryngeal mask airway (ILMA), a gum elastic bougie, a lightwand (e.g., Trachlight™), and videolaryngoscopy (VL) in some difficult airway scenarios [44,45,46,47,48]. The airway operators should know in advance that the in-place headframe may make it difficult to access the upper airway and thus have a contingency plan to handle the airway, should the need unexpectedly arise. An Allen wrench should always be available to immediately remove the crossbars of the stereotactic headframe, if needed.

A mannequin study showed that intubating with VL under an ideal situation for the Leksell front bar in situ is possible [48]. To remove just the front bar is a simpler and more streamlined approach to provide a quicker airway control in case of emergency. A mannequin study was also conducted to determine the ease of emergency airway management with a stereotactic headframe in situ [47]. Using an LMA to rescue the airway was a little quicker than using DL and VL. More than 300 DBS operations were performed by neurosurgeons at our institution in the last 10 years. The video-assisted intubating stylet (VS) technique was our preferred method for tracheal intubation in these general anesthesia cases. In our experience, the presence of the stereotactic headframe in situ has never become a barrier or a problem when using such intubation technique (Figure 7 and Appendix A).

In patients with cervical spine injuries, airway manipulation (e.g., chin lift, jaw thrust, head extension, DL) may cause segmental spine motion (rotation or displacement) and worsen the cervical instability. Flexible fiberoptic-guided nasal tracheal intubation has been reported to minimize the need of moving the cervical spine [49,50]. When compared with DL using the Macintosh blade, VL was shown to cause less upper cervical spine motion [7]. Various VL products have been compared in such restricted C-spine motion scenario, evaluating success rate, glottis visualization, intubation time, and complications [9]. VL was also found to be advantageous in facilitating nasotracheal intubation in patients with simulated C-spine injuries [51,52], and VL produced a better glottic visualization during intubation under general anesthesia with neuromuscular blockade and manual in-line stabilization. On the other hand, when dealing with a non-pathologic C-spine, the difference between the two as regards spine movement was not significant [53].

The Shikani video-assisted intubating stylet (VS) technique provides an advantage when intubating such patients with limited head/neck motion [26]. Several studies have been conducted to compare this technique with VL (using the Airway Scope^®^) in patients with cervical spine immobilization (restricted by a hard cervical collar or manual in-line stabilization) [41]. When first-attempt success, overall success, intubation time, and tissue damage were used as outcome indicators, VS and VL were comparable [41]. Inconsistent study results exist, however. Although both VL (Airway Scope^®^) and VS (StyletScope™) have high success rates in a simulated difficult airway achieved by a rigid collar, VL is faster and less likely to cause esophageal intubation [54]. VL has a higher first-attempt success rate for tracheal intubation and a shorter intubation time than VS under the condition of cervical spine manual inline stabilization [34]. On the contrary, VS produces less cervical spine motion than VL during tracheal intubation in patients with simulated cervical immobilization [33]. This finding is consistent with an earlier report that VS (Shikani stylet, Clarus Video System-Trachway^®^) was comparable to VL (Airway Scope^®^) in the success rate for tracheal intubation but provided a faster and easier intubation in patients with cervical collars [40]. Similarly, VS has been demonstrated to be a viable alternative to VL in patients with rigid cervical collars simulating a difficult airway [37]. Numerous studies have been conducted to compare various airway tools under different scenarios (Table 1). In our institution, we conduct about 600 C-spines surgeries annually, and the VS technique is used in the great majority of cases.

The cases in this report indicate that VS has a high first-attempt success rate, is smooth, and allows for a short intubating time, with no significant soft tissue injury in patients with a stereotactic headframe (Figure 6 and Figure 7). Similarly, this is also true when VS is applied in patients with a rigid cervical collar, since we observed little if any, risky movement of the cervical spine (Figure 2, Figure 3, Figure 4 and Figure 5). This finding is consistent with experiences described in our earlier clinical reports when we did not have any concerns of restricted neck motility [27,28,29,30]. Although external laryngeal manipulation might be helpful for VL in patients with C-spine immobilization [62], such maneuver is not necessary when using VS. Usually, sizable mouth opening and optimal jaw thrust are enough for maneuvering the intubating stylet into the oropharyngeal and laryngeal inlets and then obtaining sound glottis visualization. It should be noted that in some patients in which neck manipulation (e.g., extension/flexion, jaw-thrust, or laryngoscopy) cannot be performed, the epiglottis may just lie down against the posterior pharyngeal wall (an example is shown in Figure 5). In such condition, there are several strategies can be tried. Strategy 1: Pushing the intubating stylets underneath the epiglottis before directing it anteriorly. This would be easier if the patient has an omega-shaped epiglottis (e.g., Figure 8) or if there is minimal space underneath the epiglottis (e.g., Figure 9). In that case, the stylet could pass directly into the glottis under the epiglottis. Strategy 2: Maneuvering the tip of the stylet first to the right or to the left edge of the epiglottis and then sweeping the stylet backward to the midline and subsequently anteriorly into the glottis area. Strategy 3: Directing the stylet into the right or left piriform fossa and then retracting it back in the midline position. Strategy 4: Directing the tip of the stylet towards the mid post-cricoid area and then withdrawing it until the posterior glottis comes into view. Similar trouble-shooting suggestions and tips have also been previously reported for the Bonfils stylet [22].

The advantages and disadvantages of various airway management options for a patient with restricted cervical spine mobility are summarized in the Table 2. The airway operators should always remember to follow the clinical airway management guidelines when a difficult airway is anticipated. If the patient’s head/neck motility is severely restricted, and difficult tracheal intubation is expected, awake/asleep FOB under adequate topical local anesthesia can always be considered. However, this paper demonstrates that the video-assisted intubating stylet technique is an acceptable alternative airway management method in patients with restricted or confined neck motility.

## Figures and Tables

**Figure 1 healthcare-10-01688-f001:**
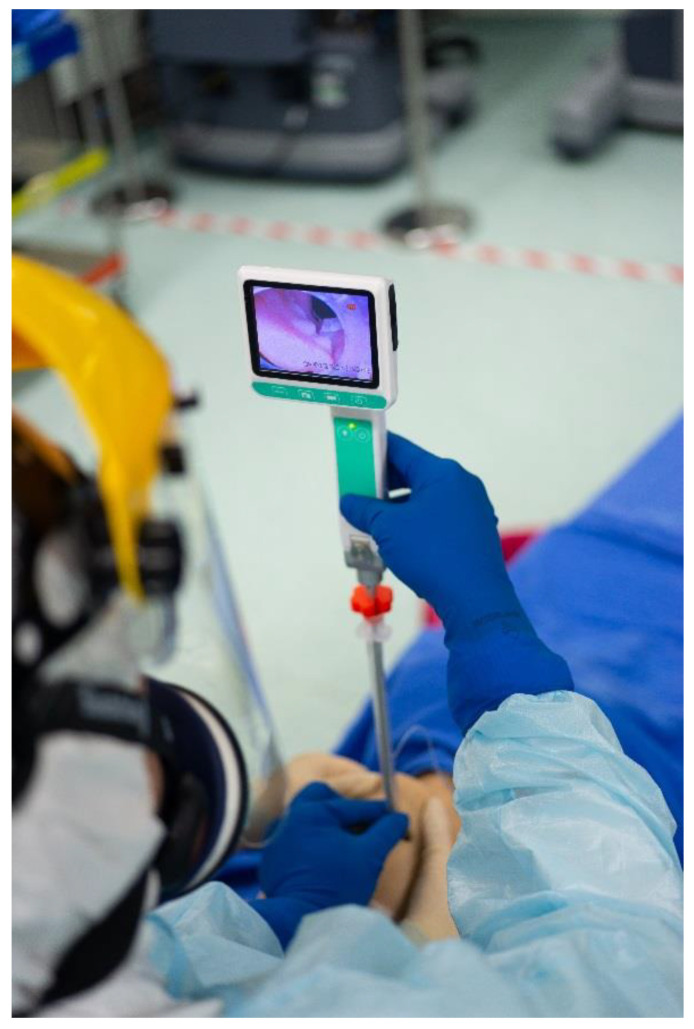
A representative demonstration of applying the video-assisted intubating stylet technique in a mannequin model. With proper mouth opening and jaw thrust assisted by a second airway operator, it is easy and smooth to complete the tracheal intubation process (also see Appendix A).

**Figure 2 healthcare-10-01688-f002:**
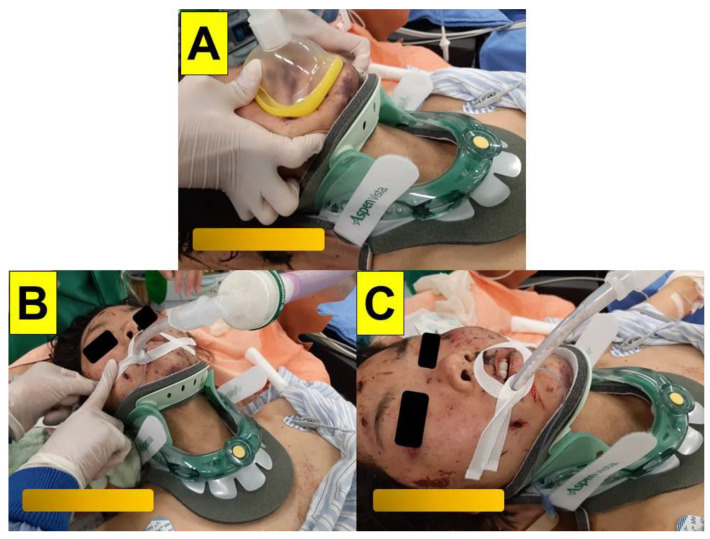
(Case 1) A 30-year-old woman with multiple trauma for emergency surgery. A rigid cervical collar was applied before the induction of anesthesia. (**A**) Face mask ventilation, (**B**,**C**) airway secured after endotracheal intubation using the video-assisted intubating stylet technique.

**Figure 3 healthcare-10-01688-f003:**
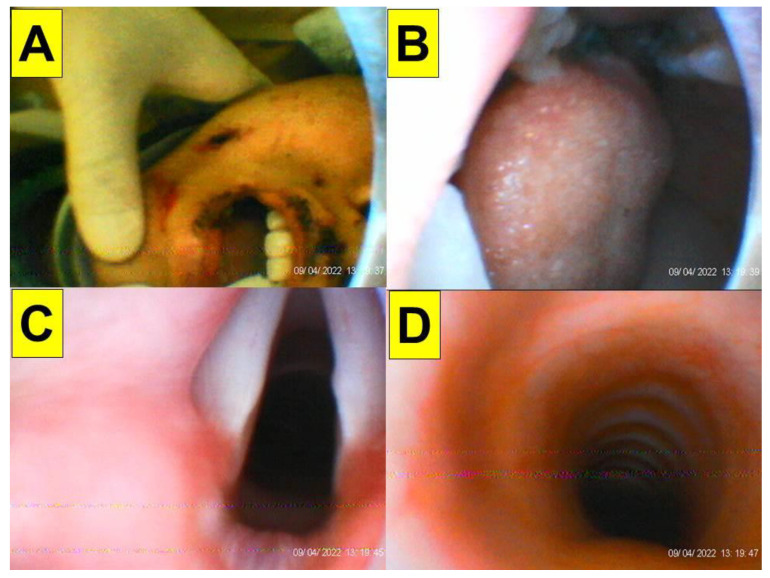
(Case 1) Close-up views during endotracheal intubation using the video-assisted intubating stylet technique. (**A**) Mouth opening, (**B**) oral space, (**C**) full glottic visualization, (**D**) entry into the trachea. The intubation process was smooth and took 7 s (from lip to trachea) to complete. A suction tube was used to remove a copious secretion before the intubation (also see Appendix A).

**Figure 4 healthcare-10-01688-f004:**
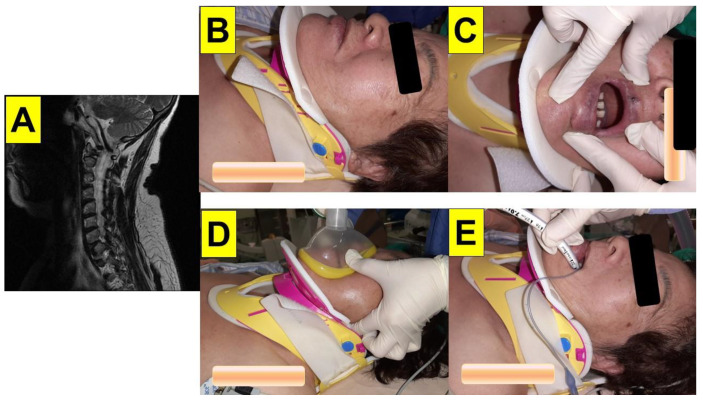
(Case 2) A 71-year-old woman with multiple trauma for emergency surgery. (**A**) Severe degenerative joint disease with neuroforaminal stenosis and spinal nerve root impingement at the 5th/6th cervical vertebrae, (**B**) a rigid cervical collar was applied before the induction of anesthesia, (**C**) relatively small mouth opening, (**D**) face mask ventilation during the induction of anesthesia, (**E**) endotracheal tube secured after the intubation using the video-assisted intubating stylet technique.

**Figure 5 healthcare-10-01688-f005:**
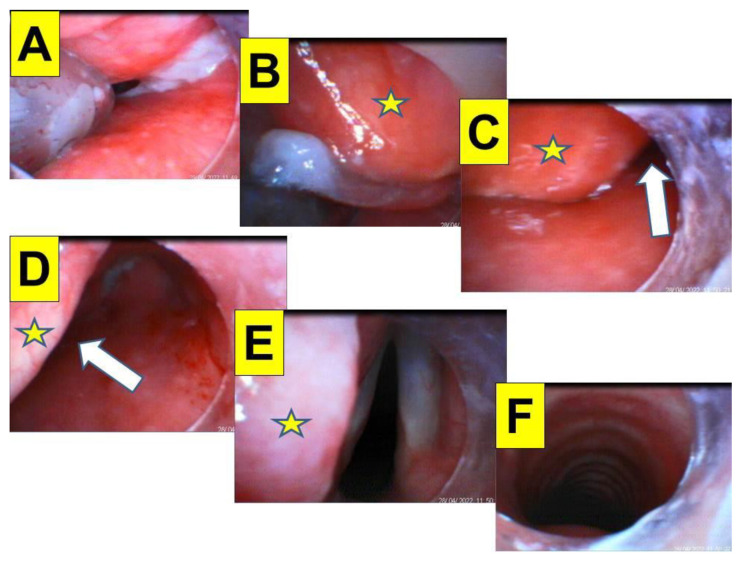
(Case 2) Close-up views during the endotracheal intubation using the video-assisted intubating stylet technique. (**A**) Before reaching the laryngeal inlet, (**B**) jaw-thrust maneuver was not applied, and the dropped epiglottis (denoted by the star) was attached to the posterior pharyngeal wall, (**C**) approaching of the intubating stylet to the glottis was facilitated by maneuvering it through the right side of the space (denoted by the white arrow), (**D**) reverting the stylet to the midline under the dropped epiglottis and directing it to the glottis, (**E**) full glottic visualization, (**F**) entry into the trachea. The intubation process was smooth and took 40 s to complete without the jaw-thrust maneuver. A suction tube was used to remove a copious secretion before the intubation (also see Appendix A).

**Figure 6 healthcare-10-01688-f006:**
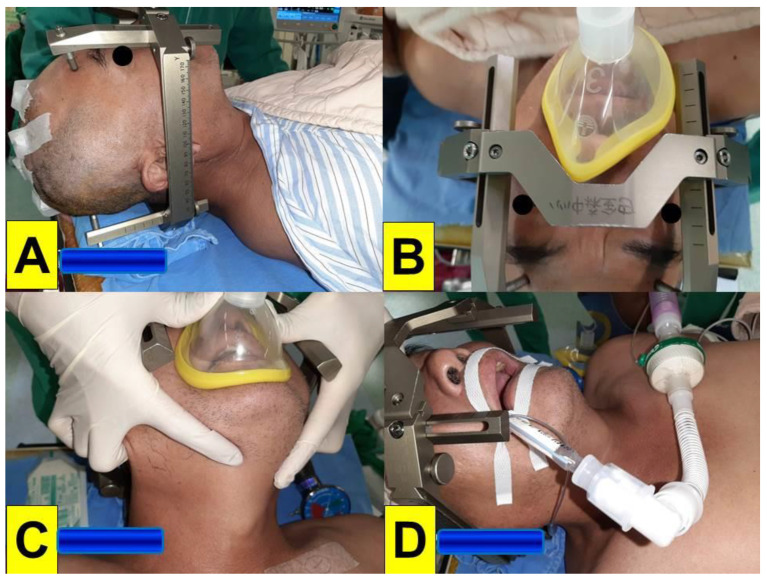
(Case 3) A 57-year-old man underwent stereotactic neurosurgery with deep-brain stimulation for Parkinson’s disease. (**A**) A stereotactic headframe was mounted before the induction of anesthesia, (**B**,**C**) face mask ventilation during the induction of anesthesia and (**D**) after endotracheal intubation. The video-assisted intubating stylet technique was used (also see Figure 7).

**Figure 7 healthcare-10-01688-f007:**
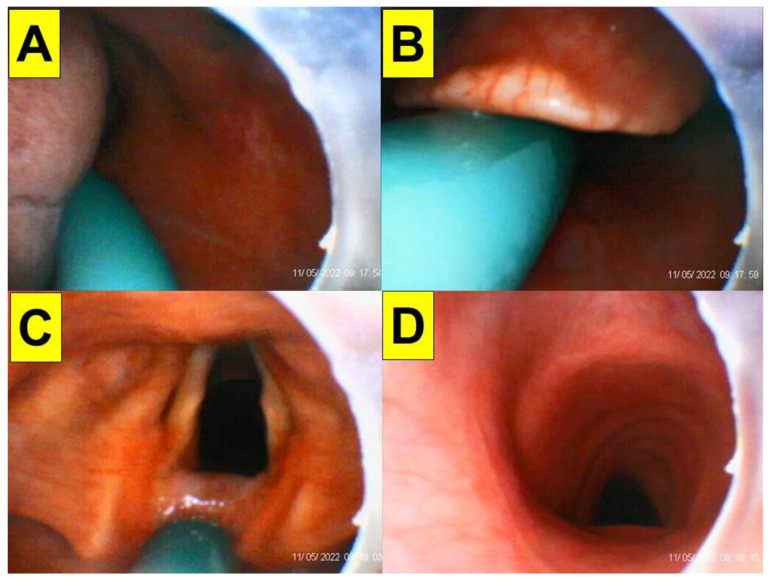
(Case 3) Close-up views during the endotracheal intubation using the video-assisted intubating stylet technique. (**A**) Oral space, (**B**) epiglottis, (**C**) full glottic visualization, (**D**) entry into the trachea. The intubation process was smooth and took 20 s to complete. A suction tube was used to remove a copious secretion before the intubation (also see Appendix A).

**Figure 8 healthcare-10-01688-f008:**
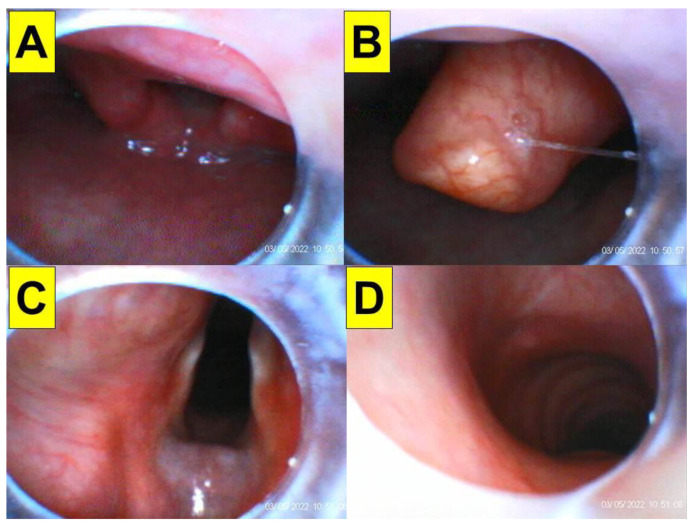
An omega-shaped epiglottis facilitates the intubating stylet technique. (**A**) Quick approach to the glottis region from the left side of the epiglottis, (**B**) view of the omega-shaped epiglottis, (**C**) full glottic visualization, (**D**) entry into the trachea. The patient was a 73-year-old man (154 cm, 65 kg, BMI 27.4 kg/m^2^, neck circumference 35 cm) who received orthopedic surgery (also see the Appendix A). It took 20 s (from the lip to the trachea) to complete the intubating process.

**Figure 9 healthcare-10-01688-f009:**
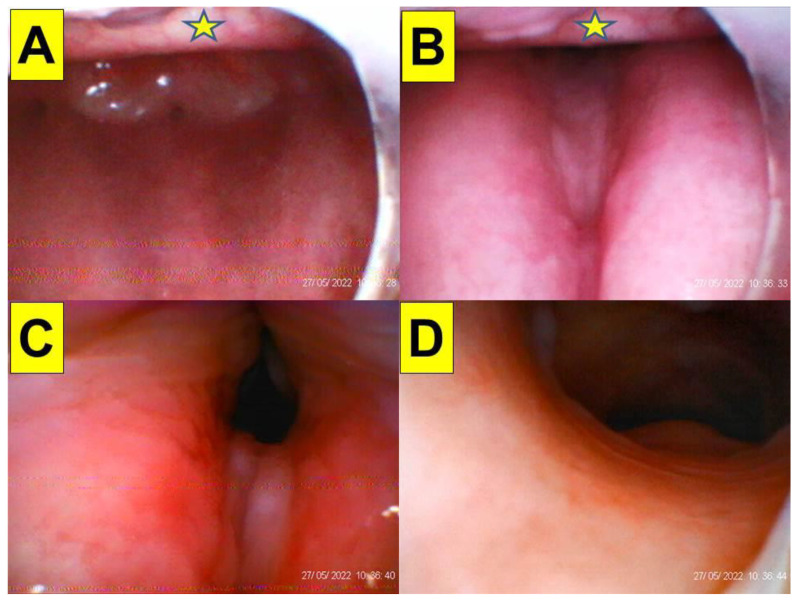
Minimal space beneath the epiglottis, enough for a midline passage of the intubating stylet. (**A**) Lifting up the flat epiglottis by the jaw-thrust maneuver, (**B**) drop of the epiglottis against the posterior pharyngeal wall without jaw-thrust effort, (**C**) full glottis visualization, (**D**) entry into the trachea. The star denoted the epiglottis in (**A**,**B**). The patient was a 74-year-old woman (152 cm, 78 kg, BMI 33.7 kg/m^2^, neck circumference 52 cm) who received orthopedic surgery (also see the Appendix A). It took 28 s (from the lip to the trachea) to complete the intubating process.

**Table 1 healthcare-10-01688-t001:** Comparison among various airway tools in restricted C-spine scenarios.

	DL	VL	FOB	VS
Using a rigid cervical collar to simulate a difficult airway in patients				
Koh [55]	−	+		
Gill [56]	−		+	
Paik [7]	−	+		
Kleine-Brueggeney [11]		+		
Kleine-Brueggeney [12]	−	+		
Bathory [57]	−	+		
Seo [51]	−	+		
Kim [58]		+		
Nam [33]		−		+
Phua [37]		−		+
Kim [40]		−		+
Using manual in-line axial cervical spine stabilization to simulate a difficult airway in patients				
Turkstra [6]	−	+		
Maharaj [59]	−	+		
Houde [31]			+	+
McElwain [60]	−	+		
Joseph [61]	−	+		
Kim [62]		+		
Robitaille [53]	−	+		
Vijayakumar [63]	−	+		
Varshney [64]		+		
Liu [65]		+		
Sinha [66]		+		
Chandy [67]		+		
Malik [68]	−	+		
Malik [69]	−	+		
Enomoto [70]	−	+		
Aoi [71]		+		
Yoo [72]	−	+		
Roh [73]	−	+		
Chan [74]				+
Turkstra [32]	−			+
Using a rigid cervical collar/manual in-line stabilization to simulate a difficult airway in mannequins				
Park [38]		−		+
Madziala [75]	−	+		
Choi [76]	−	+		
Smereka [77]	−	+		
Jain [78]	−	+		
Karczewska [79]	−	+		
Gawlowski [80]	−	+		
Pius [81]		−		+
Cervical spines injuries patients in the real world				
Yoon [34]		+		−
Xu [35]	−			+
Mahrous [36]			+	+
Wang [39]				+
Nair [82]	−	+		
Bharti [83]	−	+		
Patel [84]		+		
Shravanalakshmi [85]		+		
Jakhar [86]		+		
Kuo [52]		+		
Shih [this paper]				+

DL: conventional direct laryngoscopy; VL: all kinds of video-laryngoscopy; FOB: flexible fiberoptic bronchoscopy (used either alone or combined with other adjunctive airway tools); VS: video-assisted intubating stylet (including all optical stylets and endoscopy). +/−: + indicates better, and − indicates worse when traditional parameters were used for comparison (first-attempt and overall success rates, time to intubate, glottis visualization, subjective easiness for intubation, need for additional external maneuvers, autonomic stimulation, less complications, cervical spine motion).

**Table 2 healthcare-10-01688-t002:** Comparison of airway management modalities for patients with restricted cervical spine mobility.

	DL	VL	SGD	FOB (Awake)	VS
Need patient’s cooperation	−	−	−	+	−
Allow neurologic exam before/after intubation	−	−	−	+	−
Require wide mouth opening	+	+	−	−	−
Possible difficult laryngoscopy	+	+	−	−	−
Always better laryngeal visualization	−	+	−	+	+
Operator’s skill/experience demanded	+	−	−	+	−
High first-attempt success rate	−	+	+	+	+
High overall success rate	−	+	+	+	+
Fast intubation/easiness	−	+	+	−	+
Blood/secretion may obscure the camera view, and suction is required	−	+	−	+	+
Often require an adjunctive tool (e.g., GEB)	+	+	−	−	−
Jaw-thrust is a helpful maneuver	−	−	−	−	+
Dental/soft tissue injuries	+	+	−	−	−
Autonomic stimulation	+	+	−	−	−
Risk of pulmonary aspiration under RSI	+	+	+	−	+
Availability/affordability	+	−	+	−	−
For routine use	+	+	−	+	+
Speedy preparedness, easy maintenance	+	+	+	−	+

DL: direct laryngoscope. VL: videolaryngoscope. SGD: supra-glottic device. FOB: fiberoptic bronchoscope. VS: video-assisted intubating stylet. GEB: gum elastic bougie. RSI: rapid sequence induction/intubation.

## Data Availability

Not applicable.

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
