# Peer review of "The Use of the Shikani Video-Assisted Intubating Stylet Technique in Patients with Restricted Neck Mobility"

_healthcare, 2022, doi:10.3390/healthcare10091688_

Round 1

Reviewer 1 Report

Dear Authors,

Thank you for the opportunity to review this interesting case series on video-assisted stylet-intubation.

The manuscript provides interesting description of the video-assisted stylet-intubation in patients with restricted neck movement.  Overall the presentation of the cases is appropriate and followed by the in-depth discussion on previous studies on the subject. The discussion could be more concise and focused. Some of the aspects might be reduced and unnecessary add to the length of the discussion.   I attach my comments to the manuscript they are meant to be helpful and improve the presentation of the paper.

Introduction:

Lines 33-36 More descriptive explanation might be helpful to the reader:

Sniffing position, traditionally used during tracheal intubation, involves near-full extension of the atlanto-occipito and atlanto-axial joints and flexion of the lower cervical spine.

Austin N, Krishnamoorthy V, Dagal A. Airway management in cervical spine injury. Int J Crit Illn Inj Sci. 2014 Jan;4(1):50-6. doi: 10.4103/2229-5151.128013. PMID: 24741498; PMCID: PMC3982371

Line 37-38 this sentence might be redundant

Line 39-45 If possible, please focus more on general devices: video laryngoscopes, angulated blades, intubation ETT, or intubation stylets. You can discuss briefly benefit of each device.

Case presentation:

Line 74 It might be worth rephrasing as minimal movement of cervical spine was medically indicated

Line 75-78 First induction of anesthesia RSI? Doses? It interesting to see local practices especially when alternative technique is being described  

Line 99  “relatively small mouth opening and range of motion of the neck” It not precise description. Try to use more precise description and assessment.

Line 100 Again might need to be rephrase  as it rather a medical indication than surgical request

 Line 101 General anesthesia was induced and standard monitors were used as described above. Where is description of anesthetic technique and monitoring ?

Line 126 for readers not familiar with the procedure please explain why frame is applied before GA

Line 154  This sentence is too complicated please reduce it to indication of restricted neck extension for

Discussion:

Discussion could be more consisted and focused on VS in comparison to available technique

1.       Gold standard flexible endoscopes

2.       VL – standard and angulated

3.       DL

A general overview on available devices and it benefits.

Also discussion is not to introduced new data ( table) – this table is not necessary for the discussion.

Please shorten the discussion to a more appropriate format for a case report.

Comparison with other techniques, contraindications to this techniques?  Bleeding?  High aspiration risk? Practical tips form experience center

Author Response

Response to the Reviewer 1

The manuscript provides interesting description of the video-assisted stylet-intubation in patients with restricted neck movement. Overall the presentation of the cases is appropriate and followed by the in-depth discussion on previous studies on the subject.

The discussion could be more concise and focused. Some of the aspects might be reduced and unnecessary add to the length of the discussion. I attach my comments to the manuscript they are meant to be helpful and improve the presentation of the paper.

Thanks for your review and excellent points which are definitely very crucial, helpful and constructive. We really appreciate your expertise and professional opinions.

Introduction:

Q1: Lines 33-36 More descriptive explanation might be helpful to the reader:

“Sniffing position, traditionally used during tracheal intubation, involves near-full extension of the atlanto-occipito and atlanto-axial joints and flexion of the lower cervical spine. “ Austin N, Krishnamoorthy V, Dagal A. Airway management in cervical spine injury. Int J Crit Illn Inj Sci. 2014 Jan;4(1):50-6. doi: 10.4103/2229-5151.128013. PMID: 24741498; PMCID: PMC3982371

A1: The following sentence has been added into the text and referenced with [1] in the Introduction section. After carefully reading through, we also found this review article [1] (UW experience) is very educational and serves as a good reference. Thanks for your suggestion.

  • Sniffing position, traditionally used during tracheal intubation, involves near-full extension of the atlanto-occipito and atlanto-axial joints and flexion of the lower cervical spine
  • [1] Austin N, Krishnamoorthy V, Dagal A. Airway management in cervical spine injury. Int J Crit Illn Inj Sci. 2014;4:50-56. doi: 10.4103/2229-5151.128013.

Q2: Line 37-38 this sentence might be redundant

A2: This sentence is made to link together the direct laryngoscopy, sniffing position, traumatic C-spines, and perhaps difficult airways into the clinical scenario focused in this short report. Thanks for your comments.

Q 3: Line 39-45 If possible, please focus more on general devices: video laryngoscopes, angulated blades, intubation ETT, or intubation stylets. You can discuss briefly benefit of each device.

A3: The following statement has been added into the text:

 “The pros and cons of the available airway management options for the patient with potential cervical spine injury have been reviewed [1].

Thanks for your comments.

Case presentation:

Q4: Line 74 It might be worth rephrasing as minimal movement of cervical spine was medically indicated

A4: The sentence has been modified as follows:

Minimal movement of cervical spine during airway intervention and positioning of the patient to avoid further spine injury was made with caution.” 

Q5: Line 75-78 First induction of anesthesia RSI? Doses? It interesting to see local practices especially when alternative technique is being described. 

A5: Thanks for your comment which is an important point indeed. This patient had not received any intake for two days after the accident (She was found in the remote mountainous area and transferred to the hospital by a chopper). The fasting NPO was more than 8 hr. Although the risk of pulmonary aspiration in this patient was not outstanding, we always applied modified RSI (glycopyrrolate, midazolam, lidocaine, propofol (or ketamine/etomidate), and higher dose of rocuronium) for emergency operation. Succinylcholine was not used in multiple fractures scenario (or if hyperkalemia is a concern). For video-assisted intubating stylet technique, after adequate pre-oxygenation, we used rocuronium to paralyze the patient and no cricoid pressure was applied (if gastric content is not a major concern). The dose of rocuronium above 1 mg/kg is adequate for such intubating stylet technique. Sugammadex is always available when needed. Since intubating stylet technique is swift and smooth (and caused much less stimulation), it is particularly useful for modified RSI modality. In our institution, similar practice instruction is standing for scenarios like acute abdomen, morbid obesity, pregnant subjects, and full-stomach condition (especially confirmed by history-taking, KUB, or POC US). In addition to ramp position and NG decompression, some colleagues still feel comfortable with cricoid pressure maneuver, however.

Q6: Line 99 “relatively small mouth opening and range of motion of the neck” It not precise description. Try to use more precise description and assessment.

A6: The following sentence has been added in the text.

She presented to the operating room with a cervical collar and range of motion of the neck was limited (Figure 4). The patient moved her mouth open with limited vertical opening (2 fingers wide).

Thanks for your point.

Q7: Line 100 Again might need to be rephrase as it rather a medical indication than surgical request

A7: The following sentence has been added in the text.

Minimal movement of cervical spine during airway intervention and positioning of the patient to avoid further spine injury was made with caution.

Thank you.

Q8: Line 101 General anesthesia was induced and standard monitors were used as described above. Where is description of anesthetic technique and monitoring?

A8: The following description was added.

Standard vital-sign monitoring with electrocardiograpy (ECG), arterial blood pressure, pulse oximeter (SpO2), peripheral nerve stimulator (ToF), and end-tidal CO2 (EtCO2) were applied. General anesthesia was induced with glycopyrrolate, lidocaine, fentanyl, propofol, and rocuronium.”

Thanks for your correction.

Q9: Line 126 for readers not familiar with the procedure please explain why frame is applied before GA

A9: The following description has been added in the text.

 “The head frame for stereotactic neurosurgery is to allow precise localization of the desired target in the central nervous system. Such frame mounting is under local anesthesia. After mounting the frame in several potential clinical scenarios, magnetic resonance imaging (MRI) and stereotactical computed tomography (CT) were obtained.”

Thank you for your suggestion.

Q10: Line 154  This sentence is too complicated please reduce it to indication of restricted neck extension for

A10: This sentence has been modified as follows:

 “In this cases series, we demonstrate the use of the Shikani video-assisted intubating stylet technique in patients with restricted neck extension.

Thanks!

Discussion:

Q11: Discussion could be more consisted and focused on VS in comparison to available technique.

  1. Gold standard flexible endoscopes
  2. VL – standard and angulated
  3. DL

A general overview on available devices and it benefits.

A11: Your suggestion is reasonable, fair, and crucial. Please see our response and amendment in Q14.

Q12: Also discussion is not to introduced new data (Table 1) – this table is not necessary for the discussion.

A12: The purpose of the Table 1 is to summarize and illustrate the complex literature regarding the roles of various airways tools used for restricted cervical spine scenarios. Not only were the diverse intubating tools used, but also the models (from cadaver to real patients) were tested with. Therefore, for the readers’ convenience, we prepared the Table 1 to help the readers to go through the relevant literatures in the past decades. Meanwhile, in those clinical studies cited in the Table, the comparison among intubating tools are presented. Therefore, the Table 1 speaks for itself and contains relevant information. Thanks for your comments.

Q13: Please shorten the discussion to a more appropriate format for a case report.

A13: We do have been careful not to lengthen the Discussion section as much as possible. But since the complex issues of diverse airways tools in the scenarios of restricted cervical spine, we have to clearly state and discuss those important issues embedded in the text (e.g., advantages and disadvantages of each airway tools tested in various clinical scenarios). Therefore, if allowed, we prefer to keep all the information in the Discussion section. Fortunately, the length of the Case Presentation section is not too long. We thank you for your suggestion. 

Q14: Comparison with other techniques, contraindications to this techniques?  Bleeding? High aspiration risk? Practical tips form experience center

A14: Together with the suggestion in Q11, we propose the following revision.

We add another new Table (Table 2) to summarize the advantages and disadvantage of each category of the intubating tools used in the patient with restricted cervical spine, as you suggested. We hope this Table can facilitate the readers to catch the important information and differences between them`.

Table 2. Comparison of airway management modalities for the patient with restricted cervical spine mobility

DL

VL

SGD

FOB (awake)

VS

Need patient’s cooperation

-

-

-

+

-

Allow neurologic exam before/after intubation

-

-

-

+

-

Require wide mouth opening

+

+

-

-

-

Possible difficult laryngoscopy

+

+

-

-

-

Always better laryngeal visualization

-

+

-

+

+

Operator’s skill/experience demanded

+

-

-

+

-

High first-attempt success rate

-

+

+

+

+

High overall success rate

-

+

+

+

+

Fast intubation

-

+

+

-

+

Blood/section may obscure the camera view

-

+

-

+

+

Often require adjunctive tool (e.g., GEB)

+

+

-

-

-

Dental/soft tissue injuries

+

+

-

-

-

Autonomic stimulation

+

+

-

-

-

Risk of pulmonary aspiration under RSI

+

+

+

-

+

Availability/affordability

+

-

+

-

-

For routine use

+

+

-

+

+

Speedy preparedness, easy maintenance

+

+

+

-

+

DL: direct laryngoscope. VL: videolaryngoscope. SGD: supra-glottic device. FOB: fiberoptic bronchoscope. VS: video-assisted intubating stylet. GEB: gum elastic bougie. RSI: rapid sequence induction/intubation

The following sentence has also been added in the end of Discussion section.

The advantages and disadvantages of various airway management options for the patient with restricted cervical spine mobility are summarized in the Table 2.”

Reviewer 2 Report

This manuscript appears to be no more than an introduction of three cases using the Shikani Video-assisted intubating stylet technique.

This technique is not new, and the presentation of the three successful cases is not novel in itself.

For an academic article, an approach such as accumulating more cases or conducting scientific verification of the procedure is needed.

Author Response

Response to the Reviewer 2

Q 1: This manuscript appears to be no more than an introduction of three cases using the Shikani Video-assisted intubating stylet technique. This technique is not new, and the presentation of the three successful cases is not novel in itself.

A1: We regret that we could not convince you about our new clinical experience and communications on this particular issue. Among others, we believe the routine use of video-assisted intubating stylet technique for various kinds of restricted cervical spine mobility cases per se is kind of novelty.

Q2: For an academic article, an approach such as accumulating more cases or conducting scientific verification of the procedure is needed.

A2: We could not agree with you more. That is exactly why we present this case series before we conduct some other phases of clinical studies. Although the level of evidence of case report is not as high as those of retrospective or prospective clinical studies (or even meta-analysis), this kind of brief clinical experience communication do have its own value at the very beginning of the clinical research. Our intention is to share our own experiences regarding this special clinical application. We thank you for your comments.

Round 2

Reviewer 2 Report

We believe that the issues we pointed out in the previous report have been remained.

On the other hand, the format of the manuscript has been improved by the revisions.

Author Response

Q1: We believe that the issues we pointed out in the previous report have been remained. On the other hand, the format of the manuscript has been improved by the revisions.

A1: Thanks for your comments! For the second part of the comments, we do our best to revise the contents (and formats) accordingly. For the first part of the comments, we do have different views about the issue of the novelty of this report. Whether our short report presents any novel issue of application of the video-assisted intubating stylet technique in patients with various degrees of the restricted cervical spines mobility needs to be clarified here. Table 1 in our manuscript shows the trivial but important differences between those in literatures and ours. It is especially true in the last section of the Table 1 regarding the clinical studies conducted in the real-world situation of cervical spines surgeries (especially in the emergency situation and seriously immobilized cervical spine conditions). In such scanty literatures, one can still find major differences among the research results. Few points are emphasized as follows.

[1] Ref 36: The subjects groups included in this report were those underwent elective cervical spine surgeries. No cervical collars were applied in the study subjects. The study device is SOS, not video-assisted device. The success rate in the study group of identified difficult airways was only 94.1 %. Also the intubation time was longer (25 s). In contrast, we report the role of video-assisted intubating stylet technique in patients with cervical collar which caused significant difficulties.

[2] Ref 37: The intubation was conducted under awake status and topical local anesthetics were used. The patients did not wear collar to stabilize the unstable cervical spines. The SOS group (instead of our VS tool) shows 53 s of the intubating time and 10% of the first-attempt success rate. In contrast, our presenting cases were under routine general anesthesia induction. And the patients wore cervical collar which created certain degrees of difficulties to intubate. We used video-assisted intubating stylet instead of SOS.

[3] Ref 40: This study included patients undergoing elective cervical spines surgeries. No cervical collar or other immobilization maneuvers were involved. Interestingly, SOS was better than Trachway in some comparisons. The performance comparators in this study (first-attempt success rate, intubation time, glottis visualization, etc) are different from ours.

We not only vividly demonstrate the intubating process in patients with restricted cervical spine mobility conditions, but also show indicate some special tips for video-assisted intubating stylet technique in this patients population. One more important point we would like to emphasize is that our experiences with VS on airway management in patients undergoing deep brain stimulation surgery and mounted with stereotactic headframe is unique.

All together, we believe it is not difficult to identify the novelty in our short clinical report. And we strongly believe that our reporting clinical experiences are new and of value to the relevant academic communities. Thanks again for your efforts to allow us to clarify our position.